

**Twin-plate ice nucleation assay (TINA) with infrared detection for**
**high-throughput droplet freezing experiments with biological ice**
**nuclei in laboratory and field samples**
**Anna T. Kunert[1], Mark Lamneck[2], Frank Helleis[2], Mira L. Pöhlker[1], Ulrich Pöschl[1], and**
**Janine Fröhlich-Nowoisky[1,*]**
[1]Multiphase Chemistry Department, Max Planck Institute for Chemistry, 55128 Mainz,
Germany
[2]Instrument Development and Electronics, Max Planck Institute for Chemistry, 55128 Mainz,
Germany
*Correspondence to:* Janine Fröhlich-Nowoisky (j.frohlich@mpic.de)





**Abstract.** For efficient analysis and characterization of biological ice nuclei under immersion
freezing conditions, we developed a Twin-plate Ice Nucleation Assay (TINA) for high-
throughput droplet freezing experiments, in which the temperature gradient and freezing of
each droplet is tracked by an infrared detector. In the fully automated setup, a couple of
independently cooled aluminum blocks carrying two 96-well plates and two 384-well plates,
respectively, are available to study ice nucleation and freezing events simultaneously in
hundreds of microliter range droplets (0.1-40 µL). A cooling system with two refrigerant
circulation loops is used for high-precision temperature control (deviations <0.5 K), enabling
measurements over a wide range of temperatures (~233-270 K) at variable cooling rates (up to
10 K min$^{-1}$ and more).
The TINA instrument was tested and characterized in experiments with bacterial and
fungal ice nuclei (IN) from *Pseudomonas syringae* (Snomax®) and *Mortierella alpina*,
exhibiting freezing curves in good agreement with literature data. Moreover, TINA was applied
to investigate the influence of chemical processing on the activity of biological IN, in particular
the effects of oxidation and nitration reactions. Upon exposure of Snomax® samples to $O_3$ and
$NO_2$, the concentration of IN active around 269 K (-4 °C, "class A") decreased by about two
orders of magnitude, while the concentration of IN active around 265 K (-8 °C, "class C")
decreased by about one order of magnitude. Furthermore, TINA was used to study aqueous
extracts of atmospheric aerosols, simultaneously investigating a multitude of samples that were
treated in different ways to distinguish different kinds of IN. For example, heat treatment and
filtration experiments indicated that highly efficient biological IN were smaller than 0.1 µm,
and many IN active between 263 K (-10 °C) and 256 K (-14 °C) were heat-resistant and larger
than 5 µm. The results confirm that TINA is suitable for high-throughput experiments and
efficient analysis of biological IN in laboratory and field samples.





## 1 Introduction

Clouds and aerosols still contribute the largest uncertainty to the evaluation of the Earth's changing energy budget (Boucher et al., 2013). Thus, the understanding of the contribution of atmospheric aerosols in cloud processes is of fundamental importance. Atmospheric ice nucleation is essential for cloud glaciation and precipitation, thereby influencing the hydrological cycle and climate. Ice can be formed via homogeneous nucleation in liquid water droplets or heterogeneous nucleation triggered by particles serving as atmospheric ice nuclei (IN) (Pruppacher and Klett, 1997).

A wide range of droplet freezing assays and instruments have been developed and applied for the analysis of IN in immersion freezing experiments (e.g., Budke and Koop, 2015; Fröhlich-Nowoisky et al., 2015; Häusler et al., 2018; Murray et al., 2010; O'Sullivan et al., 2014; Stopelli et al., 2014; Tobo, 2016; Vali, 1971b, 1971a; Whale et al., 2015; Wright and Petters, 2013; Zaragotas et al., 2016). Most of the available assays and instruments, however, are limited to the investigation of small droplet numbers and use optical detection systems in the UV/Vis wavelength range. As shown by Zaragotas et al. (2016), however, infrared (IR) detector enable improved detection of droplet freezing.

Upon the phase change of water from liquid to solid, latent heat is released resulting in a sudden temperature change of the droplet, which can be detected by infrared (IR) video thermography. In 1995, Ceccardi et al. (1995) used IR video thermography as a new technique to non-destructively study ice formation on plants by visualizing the changes in surface temperature. Wisniewski et al. (1997) evaluated the IR video thermography under controlled conditions and determined it as an excellent method for directly observing ice nucleation and propagation in plants. Since then, IR video thermography was used in a range of studies investigating freezing in plants (e.g., Ball et al., 2002; Carter et al., 1999; Charrier et al., 2017; Fuller and Wisniewski, 1998; Hacker and Neuner, 2007; Pearce and Fuller, 2001; Sekozawa et al., 2004; Stier et al., 2003; Wisniewski et al., 2008; Workmaster, 1999). Further applications of IR video thermography are investigations of cold thermal stress in insects (Gallego et al., 2016), monitoring of freeze drying processes (Emteborg et al., 2014), detection of ice in wind turbine blades (Gómez Muñoz et al., 2016) and helicopter rotor blades (Hansman and Dershowitz, 1994). Freezing of single water droplets in an acoustic levitator has also been successfully observed by IR video thermography (Bauerecker et al., 2008).

Here, we introduce a Twin-plate Ice Nucleation Assay (TINA) for high-throughput droplet freezing experiments in which the temperature gradient and freezing of each droplet is tracked by an infrared detector. In the fully automated setup, a couple of independently cooled



aluminum blocks carrying two 96-well and two 384-well plastic plates, respectively, are
available to study ice nucleation and freezing events in nearly 1000 microliter range droplets
simultaneously. A cooling system with two refrigerant circulation loops is used for high-
precision temperature control (deviations <0.5 K), enabling measurements over a wide range
of temperatures (~233-270 K (~ -50 °C to -3 °C)) at variable cooling rates (up to 10 K min$^{-1}$
and more). The instrument was developed in the course of the INUIT project over the last three
years, in which it has been presented and discussed at several conferences and workshops
(Kunert et al., 2016a, 2016b, 2017a, 2017b, 2018). We use the bacterial IN Snomax® and the
IN-active fungus *Mortierella alpina* as biological test substances to investigate heterogeneous
ice nucleation. Moreover, TINA is applied to investigate the effect of $O_3$ and $NO_2$ exposure on
the IN activity of Snomax®. Furthermore, aqueous extracts of atmospheric aerosols are treated
in different ways and are analyzed for different kinds of IN. Very recently, a similar approach
for droplet freezing experiments with IR detection has been presented by Harrison et al. (2018),
investigating K-feldspar, NX-illite, and atmospheric aerosol samples.

**2 Experimental setup**
**2.1 Technical details**
The core of the Twin-plate Ice Nucleation Assay (TINA) are two independently cooled,
customized sample holder aluminum blocks, which have been shaped for multiwell plates with
96 and 384 wells, respectively. In each cooling block, two multiwell plates can be analyzed
simultaneously. The maximal droplet volume in the 96-well block is 250 µL, and the minimal
droplet volume is 0.1 µL, which is the limit of our liquid handling station (epMotion ep5073,
Eppendorf, Hamburg, Germany). As shown in Fig. 1, the design of the two sample holder
blocks is basically identical, but the detailed construction varies slightly. Both blocks consist
of two parts, a trough and a cap, which are screwed together and sealed with an O-ring. But, for
the 96-well block (Fig. 1a), the trough is at the bottom (Fig. 1c) and the cap is at the top (Fig.
1b), whereas, for the 384-well block (Fig. 1d), the trough is at the top (Fig. 1e) and the cap is
at the bottom (Fig. 1f). Two openings with Swagelok adapters for cooling liquid are placed next
to each other, and the cooling liquid flows in a small passage around an elevation in the middle
of the trough.
The customized sample holder blocks are cooled with a silicon-based cooling liquid
(SilOil M80.055.03, Peter Huber Kältemaschinenbau AG, Offenburg, Germany) tempered by
an external high-performance refrigeration bath circulator (CC-508 with Pilot ONE, Peter
Huber Kältemaschinenbau AG), which can supply temperatures down to 218 K (-55 °C). Both



sample holder blocks can be operated in parallel and independently from each other due to the
usage of two self-developed mixing valves and cooling loops (Fig. 2). This allows either the
cooling of two different droplet freezing assays at the same time, or the observation of 960
droplets in one experiment. The mixing of a cold and a warm loop of cooling liquid for each
block enables a fast and precise adjustment of the sample holder block temperatures without
being dependent on the cooling rate of the refrigeration bath circulator itself. In each
experiment, the refrigeration bath circulator is cooled down 5 K below the coldest temperature,
which is projected for the experiment, while no mixing of warm and cold cooling liquid occurs.
By changing the position of the mixing valves for a defined period of time, cold and warm
cooling liquids are mixed together, so that the desired temperatures within the two blocks are
reached. Two pumps (VPP-655 PWM Single Version, Alphacool International GmbH,
Braunschweig, Germany) ensure the continuous circulation of cooling liquid through each
block independently from the position of the mixing valves. Figure 3 is a schematic illustration
of the overall setup of TINA.

**2.2 Temperature control and calibration**
Within each sample holder block, the temperature is measured with two temperature sensors, a
NTC thermistor in the cooling liquid stream (TH-44033, resistance: 2255 Ω/298 K,
interchangeability: ±0.1 K, Omega Engineering GmbH, Deckenpfronn, Germany) and a
customized NTC thermistor (10K3MRBD1, resistance: 10000 Ω/298 K, interchangeability:
±0.2 K, TE Connectivity Company, Galway, Ireland) in the elevation. Another NTC thermistor
(10K3MRBD1, resistance: 10000 Ω/298 K, interchangeability: ±0.2 K, TE Connectivity
Company) monitors the temperature behind each mixing valve. Temperature control within the
entire system is achieved by a self-developed microcontroller-based electronic system. The
analog input unit is equipped with a 24 Bit Low Noise Delta-Sigma ADC (ADS1256), which
assures the required accuracy to process the resolution of the used thermistors. All thermistors
had been calibrated with a reference thermometer (2180A, resolution: 0.01 K, maximum system
error: ± 0.08 K at 223 K and ± 0.07 K at 273 K, Fluke Deutschland GmbH, Glottertal,
Germany). Therefore, all thermistors were bound together with a PT100 sensor of the reference
thermometer, and the bundle was placed inside a brass cylinder filled with cooling liquid. The
cylinder was placed inside the cooling bath of the refrigeration bath circulator. The temperature
within the bath was cooled down from 303.2 K to 218.2 K (30.0 °C to -55.0 °C) in 5 K steps
and raised again from 220.7 K to 300.7 K (-52.5 °C to 27.5 °C) in 5 K steps. Each step was kept
for 30 min to equilibrate the temperature, while the resistance of all thermistors and the





temperature measured by the reference thermometer were monitored. For the conversion of the
measured resistance of the thermistors into temperature, cubic spline interpolation was used.

**2.3 Infrared video thermography**
Droplet freezing is determined by a distinct detection system, where the temperature gradient
of each single droplet is tracked by infrared cameras (Seek Thermal Compact XR, Seek
Thermal Inc., Santa Barbara, CA, USA) coupled to a self-written software. This concept
enables a doubtless determination of freezing events because freezing of supercooled liquid
releases energy, which leads to an abrupt rise in the detected temperature of the observed
droplet, as discussed earlier (Sect. 1). This detection system uses the IR video thermography
only to determine freezing events, while the proper temperature is monitored by thermistors. In
contrast, Zaragotas et al. (2016) used infrared camera, which was calibrated only once by the
company, to measure the accurate temperature of each droplet. Figure 4 is a sequence of
infrared camera images showing 384 droplets during cooling and freezing (red circles).
Software analysis uses a grid of 96 and 384 points, respectively, where the grid point is set to
the center of each well enabling to fit the dimensions of each plate under different perspective
angles. The temperature gradient is tracked for each well during the experiment. A self-written
algorithm detects a local maximum shortly followed by a local minimum in the derivative of
the temperature profile, which is caused by the release of latent heat during freezing. The
software exports the data for each droplet in CSV format.

**2.4 Data analysis**
Assuming ice nucleation as a time-independent (singular) process, the concentration of IN ($n_{\mathrm{m}}$)
active at a certain temperature ($T$) per unit mass of material is given by Eq. (1) (Vali, 1971a).
$$n_{\mathrm{m}}(T) = \frac{-\ln(1 - f_{\mathrm{ice}})}{V_{\mathrm{drop}}} \cdot \frac{d}{m} \qquad (1)$$

where $f_{\mathrm{ice}}$ is the fraction of frozen droplets at a particular temperature, $V_{\mathrm{drop}}$ the droplet volume,
$m$ the mass of the particles in the initial suspension, and $d$ the dilution factor of the droplets
relative to $m$. To simplify data analysis, freezing events were merged in 0.1 K bins.

**3. Freezing experiments**
The fully automated TINA setup was tested and characterized for immersion freezing
experiments with pure water droplets, as well as Snomax® and IN filtrate of the fungus
*Mortierella alpina* as biological reference substances. Moreover, TINA was used to study the





effect of O₃ and NO₂ exposure on the IN activity of Snomax®. Furthermore, TINA was applied
to atmospheric aerosol samples.

**3.1 Pure water**
Pure water was obtained from a Barnstead™ GenPure™ xCAD Plus water purification system
(Thermo Scientific, Braunschweig, Germany). The water was autoclaved at 394 K (121 °C) for
20 min, filtered three times through a sterile 0.1 µm pore diameter sterile polyethersulfone
(PES) vacuum filter unit (VWR International, Radnor, PA, USA), and autoclaved again.

For background measurements, 3 µL aliquots of autoclaved and filtered pure water were

pipetted into 96-well plates (Axon Labortechnik, Kaiserslautern, Germany) and 384-well plates
(Eppendorf), respectively, by a liquid handling station. Therefore, four (96-well plate) and eight
(384-well plate) different water samples were pipetted column-wise distributed into the plates.
In total, six columns per sample were apportioned over the two twin-plates, i.e., 48 droplets per
sample in 96-well plates, and 96 droplets per sample in 384-well plates. The plates were placed
in the sample holder blocks and were cooled down quickly to 273 K (0 °C) and, as soon as the
temperature was stable for one minute, in a continuous cooling rate of 1 K min⁻¹ further down
to 238 K (-35 °C).

As the phase transition from liquid water to ice is kinetically hindered, supercooled

water can stay liquid at temperatures down to 235 K (-38 °C), where homogeneous ice
nucleation takes place. This is only true for nanometer-sized droplets because the freezing
temperature is dependent on droplet volume and cooling rate, and the classical nucleation
theory predicts a homogeneous freezing temperature of about 240 K (-33 °C) for microliter
volume droplets using a cooling rate of 1 K min⁻¹ (Fornea et al., 2009; Murray et al., 2010;
Pruppacher and Klett, 1997; Tobo, 2016). However, several studies have reported average
freezing temperatures for microliter volume droplets of pure water at significantly warmer
subfreezing temperatures due to possible artifacts (e.g., Conen et al., 2011; Fröhlich-Nowoisky
et al., 2015; Hill et al., 2016; Whale et al., 2015). To our knowledge, only two studies reported
an average homogeneous freezing temperature of 240 K (-33 °C) for microliter volume droplets
and a cooling rate of 1 K min⁻¹, using hydrophobic surfaces as contact area for the droplets
(Fornea et al., 2009; Tobo, 2016). Providing microliter droplets free of suspended IN and
surfaces free of contaminants is difficult, so that the temperature limit below which freezing
cannot be traced back to heterogeneous IN needs to be determined individually for each setup.

Our results show that most pure water droplets froze around 247 K (-26 °C) in 96-well

plates (Fig. 5a) and around 244 K (-29 °C) in 384-well plates (Fig. 5b), respectively. This



discrepancy can have different explanations. First, the 96-well plates are obtained from a
different manufacturer than the 384-well plates. Second, the different well shape leads to an
altered shape of the droplet, which could influence its freezing abilities at very low
temperatures. All in all, these freezing temperatures are substantially above the expected
temperatures for homogeneous nucleation of microliter droplets, but they are in accord with the
results of Whale et al. (2015).

**214   3.2 Biological reference materials**

The performance of TINA was further assessed using Snomax® as a bacterial IN-active
reference substance (e.g., Budke and Koop, 2015; Hartmann et al., 2013; Möhler et al., 2008;
Turner et al., 1990; Ward and DeMott, 1989) and IN filtrate of the well-studied IN fungus
*Mortierella alpina* (Fröhlich-Nowoisky et al., 2015; Pummer et al., 2015).

Snomax® was obtained from SMI Snow Makers AG (Thun, Switzerland), and a stock

solution was prepared in pure water with an initial mass concentration of 1 mg mL$^{-1}$. This
suspension was then serially diluted 10-fold with pure water by the liquid handling station. The
resulting Snomax® concentrations varied between 1 mg mL$^{-1}$ and 0.1 ng mL$^{-1}$, equivalent to a
total mass of Snomax® between 3 µg and 0.3 pg, respectively, per 3 µL droplet.

Each dilution was pipetted column-wise distributed over the twin-plates as described

before in 96 droplets into 384-well plates by the liquid handling station. Two plates at a time
were placed inside the 384-well sample holder block, and the plates were cooled down quickly
to 273 K (0 °C), and, as soon as the temperature was stable for one minute, in a continuous
cooling rate of 1 K min$^{-1}$ further down to 253 K (-20 °C).

Three independent experiments with Snomax® show reproducible results (Fig. S1).

Therefore, droplets of the same dilution were added to a total droplet number of 288 (Fig. 6a).
The data show two strong increases in the cumulative number of IN, one around 269 K (-4 °C)
and one around 265 K (-8 °C), interrupted by a slightly increasing plateau between 268 K and
266 K (-5 °C and -7 °C). These differences result from three different classes of IN with
different activation temperatures as described by Turner et al. (1990). Based on this
classification, the Snomax® sample contains a large number of class A and C IN, but only a few
IN of class B. These findings are in accordance with the results of Budke and Koop (2015).
Below 259 K (-14 °C), a flat plateau arises where no IN are active.

The analysis of different IN active within a wide temperature range was only possible

due to the measurement of a dilution series. TINA enables the simultaneous measurement of
such a dilution series with high statistics in a short period of time.



*Mortierella alpina* 13A was grown on full-strength PDA (VWR International GmbH,
Darmstadt, Germany) at 269 K for 7 months. Fungal IN filtrate was prepared as described
previously (Fröhlich-Nowoisky et al., 2015; Pummer et al., 2015) and contained IN from spores
and mycelial surfaces. It was serially diluted 10-fold with pure water by the liquid handling
station. The experiment was performed as described above.
For test measurements with fungal IN, IN filtrate of three different culture plates from
*Mortierella alpina* 13A was measured, and the results were reproducible (Fig. S2). The
cumulative number of IN per gram mycelium only varies between one order of magnitude,
which is a good achievement for a biological sample, and droplets of the same dilutions were
added to a total droplet number of 288 (Fig. 6b). The cumulative number of IN and the initial
freezing temperature of 267.7 K (-5.5 °C) are in good agreement with the literature (Fröhlich-
Nowoisky et al., 2015; Pummer et al., 2015).

**3.3 Ozonized and nitrated samples**
To study the effect of $O_3$ and $NO_2$ exposure on the IN activity of Snomax®, an aliquot of 1 mL
of a 1 mg mL$^{-1}$ suspension of Snomax® in pure water was exposed in liquid phase to gases with
or without $O_3$ and $NO_2$ as described in Liu et al. (2017). A dilution series of the treated samples
was measured as described for the Snomax® reference measurements.
The results show that high concentrations of $O_3$ and $NO_2$ reduce the concentration of IN
active around 269 K (-4 °C, "class A") about two orders of magnitude, while the concentration
of IN active around 265 K (-8 °C, "class C") decreased by about one order of magnitude (Fig.
7). The cumulative number of IN per unit mass is slightly reduced by the exposure to synthetic
air, which can be explained by a small loss of IN within the system during exposure.
Snomax® contains IN proteins of the bacterium *Pseudomonas syringae*. Attard et al.,
(2012) found no significant or only weak effects of exposure to ~100 ppb $O_3$ and ~100 ppb $NO_2$
on the IN activity of two strains of *P. syringae*, and a variable response of a third strain,
suggesting a strain-specific response.

**3.4 Air filter samples**
Total suspended particle samples were collected onto 150 mm glass fiber filters (Type MN
85/90, Macherey-Nagel GmbH, Düren, Germany) using a high-volume sampler (DHA-80,
Digitel Elektronik AG, Hegnau, Switzerland) operated at 1000 L min$^{-1}$, which was placed at
the roof of the Max Planck Institute for Chemistry (MPIC, Mainz, Germany). There, a mix of
urban and rural continental boundary layer air can be sampled in central Europe. The filter was



taken in April 2018, and the sampling period was seven days, corresponding to a total air
volume of approximately 10,000 m³. Filters were pre-baked at 603 K (330 °C) for 10 h to
remove any biological material, and blank samples were taken to detect possible
contaminations. All filters were packed in pre-baked aluminum bags, and loaded filters were
stored at 193 K (-80 °C) until analysis.

Filters were cut with a sterilized scissor into aliquots (~1/16), and the exact percentage

was determined gravimetrically. For reproducibility, aerosol and blank filter sample aliquots of
each filter were extracted. Each filter sample aliquot was transferred into a sterile 50 mL tube
(Greiner Bio One, Kremsmünster, Austria), and 10 mL of pure water was added. The tubes
were shaken horizontally at 200 rpm for 15 min. Afterwards, the filter was removed, and the
aqueous extract was tested for IN activity. To further characterize the IN, the effects of filtration
and heat treatment were investigated. Therefore, aliquots of the extract were treated as follows:
(i) 1 h at 371 K (98 °C), (ii) filtration through a 5 µm pore diameter filter (Acrodisc, PES, Pall,
Germany), (iii) filtration through a 5 µm and a 0.1 µm pore diameter filter (Acrodisc).

Each solution (96 aliquots of 3 µL) was pipetted column-wise into 384-well plates by

the liquid handling station. The plates were cooled down quickly to 273 K (0 °C), and, as soon
as the temperature was stable for one minute, in a continuous cooling rate of 1 K min⁻¹ further
down to 243 K (-30 °C).

Each solution of the two aliquots of each filter was measured separately, and droplets

of the same solution were added to a total droplet number of 192 (2 x 96 droplets) (Fig. S3).
For better clearness, data of different dilutions were averaged for each treatment (Fig. 8).

The untreated filter extract showed IN activity at relatively warm subfreezing

temperatures with an initial freezing temperature of 267 K (-6 °C). The concentration of IN
active at temperatures above 263 K (-10 °C) was about $10^{-3}$ L⁻¹, but heat treatment led to a loss
of IN activity above 263 K (-10 °C), which indicates the presence of biological IN. The
concentration of IN between 263 K (-10 °C) and 255 K (-18 °C) increased continuously about
two orders of magnitude and in a sudden increase another two orders between 255 K (-18 °C)
and 254 K (-19 °C). A maximum IN concentration of $10^2$ L⁻¹ was reached around 250 K (-23
°C), but heat treatment reduced the maximum IN concentration to $10^1$ L⁻¹ at 250 K (-23 °C).
Filtration experiments did not affect the initial freezing temperature, but the concentration of
biological IN was decreased about half an order of magnitude, at which the 0.1 µm filtration
showed a slightly bigger effect. The concentration of IN active between 263 K (-10 °C) and
256 K (-17 °C) decreased about two orders of magnitude upon 5 µm filtration, and 0.1 µm
filtration reduced the IN concentration slightly more. The maximum IN concentration of $10^2$ L⁻





[1] around 250 K (-23 °C) was not affected upon filtration. The results suggest that there were
highly efficient biological IN smaller than 0.1 μm and other biological IN or agglomerates of
the same biological IN with different sizes. Moreover, the findings show that many IN active
between 263 K (-10 °C) and 256 K (-17 °C) were larger than 5 μm, whereas IN active between
256 K (-17 °C) and 250 K (-23 °C) were smaller than 0.1 μm. Most of the IN active between
263 K (-10 °C) and 259 K (-14 °C) were heat-resistant.

The results of both, the untreated and the heated filter extracts, showed a few outliers.

These were probably caused by single large particles or aggregates larger than 5 μm, which
were statistically distributed over the different dilutions. These particles nucleated at warmer
subfreezing temperatures than the other IN within a dilution, so they are overestimated due to
their efficient nucleation. This hypothesis is supported by the fact that the filtered filter extracts
did not contain any outliers.

**4 Conclusions**
The new high-throughput droplet freezing assay TINA was introduced to study heterogeneous
ice nucleation of microliter range droplets in the immersion mode. TINA provides the analysis
of 960 droplets simultaneously or 192 and 768 droplets in two independent experiments at the
same time, enabling the analysis of many samples with high statistics in a short period of time.
Moreover, an infrared camera-based detection system allows to reliably determine droplet
freezing. The setup was tested with Snomax® as bacterial IN, and IN filtrate of *Mortierella*
*alpina* as fungal IN. For these reference materials, both, the initial freezing temperature and the
cumulative number of IN per gram mycelium, were in accordance with the literature, which
demonstrates the functionality of the new setup.

TINA was applied to study the effect of $O_3$ and $NO_2$ exposure on the IN activity of

Snomax®, where high concentrations of $O_3$ and $NO_2$ reduced the IN activity significantly.
Atmospheric aerosol samples from Mainz (Germany) were analyzed for IN activity to show the
applicability of TINA for field samples. Here, we found highly efficient biological IN smaller
than 0.1 μm and heat-resistant IN larger than 5 μm. The results confirm that TINA is suitable
for high-throughput experiments and efficient analysis of biological IN in laboratory and field
samples.



*Author Contributions.* A.T.K., M.L., and F.H. developed the instrument. A.T.K., U.P., J.F.-N.
conceived and designed the experiments. A.T.K. performed the experiments. M.L.P. wrote the
code to process the data. All authors discussed the data and contributed to the writing of the
manuscript.

*Competing Interests.* The authors declare that they have no conflict of interest.

*Acknowledgements.* The authors thank C. Gurk, T. Klimach, F. Kunz and the workshop team
for supporting the experimental setup, N. M. Kropf, C. Krevert, I. Maurus, G. Kopper, and P.
Yordanova for technical support, and H. Grothe, T. Koop, N. Lang-Yona, and J. F. Scheel for
helpful discussions. The Max Planck Society (MPG) and the Ice Nuclei research Unit of the
Deutsche Forschungsgemeinschaft (DFG FR 3641/1-2, FOR 1525 INUIT) are acknowledged
for financial support.





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




(a) 96-well sample holder block      (d) 384-well sample holder block

(b) 96-well top (flipped)      (e) 384-well top (flipped)

(c) 96-well bottom      (f) 384-well bottom

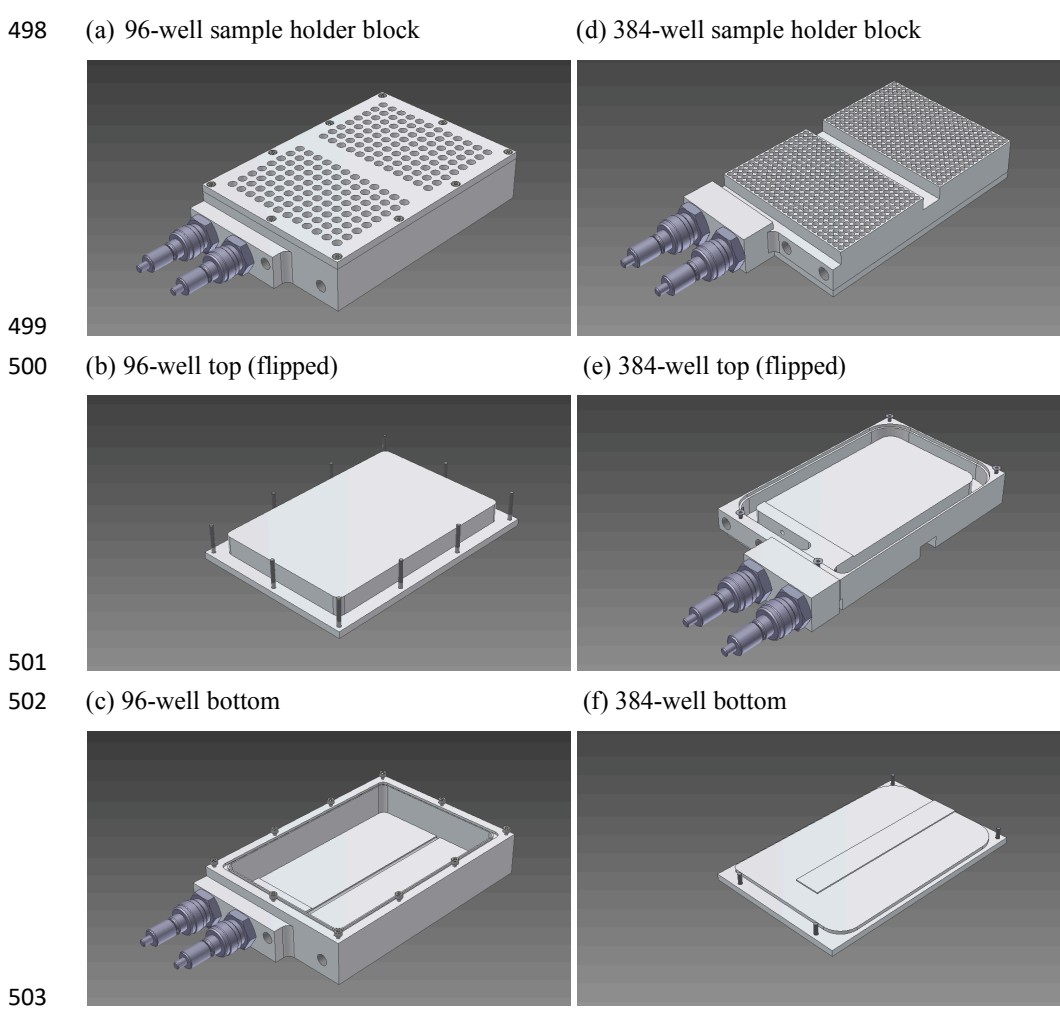


**Figure 1.** Sample holder and cooling blocks of the Twin-plate Ice Nucleation Assay (TINA)
with **(a-c)** 96-well plates and **(d-f)** 384-well plates (CAD drawings).





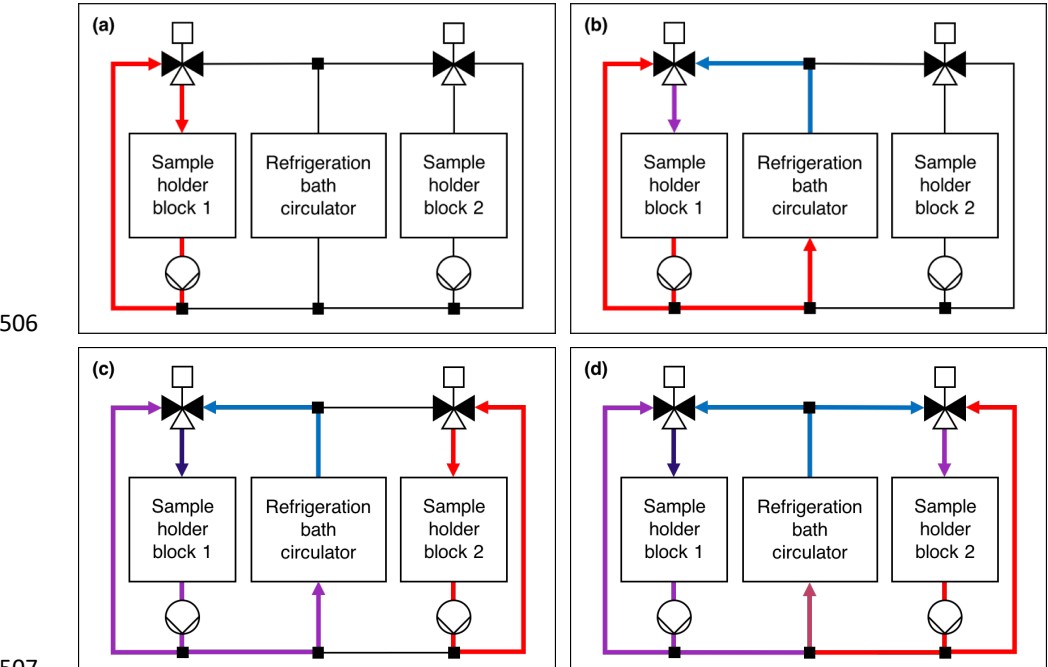

**Figure 2.** Cooling system layout and operating principle of the Twin-plate Ice Nucleation Assay (TINA). **(a)** Cooling liquid is pumped in warm cooling loop of sample holder block 1 without connection to colder cooling liquid provided by refrigeration bath circulator. **(b)** Mixing valve is opened for both, warm cooling liquid of warm cooling loop and cold cooling liquid of refrigeration bath circulator. Position of mixing valve defines temperature within sample holder block 1. **(c)** Sample holder block 1 is cooled further down, while cooling liquid is pumped in warm cooling loop of sample holder block 2. **(d)** Sample holder block 2 can be run in parallel independently from the temperature in sample holder block 1.





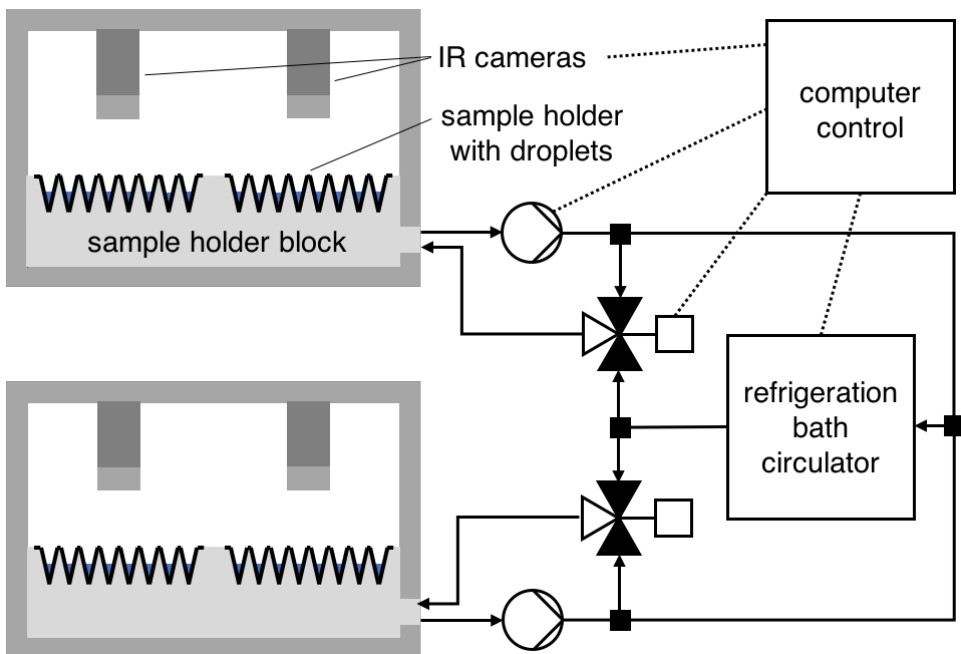

516

**Figure 3.** Schematic illustration of the overall setup: sample holder blocks, sample holders with droplets, IR cameras, cooling system with refrigeration bath circulator, pumps and mixing valves, computer control.



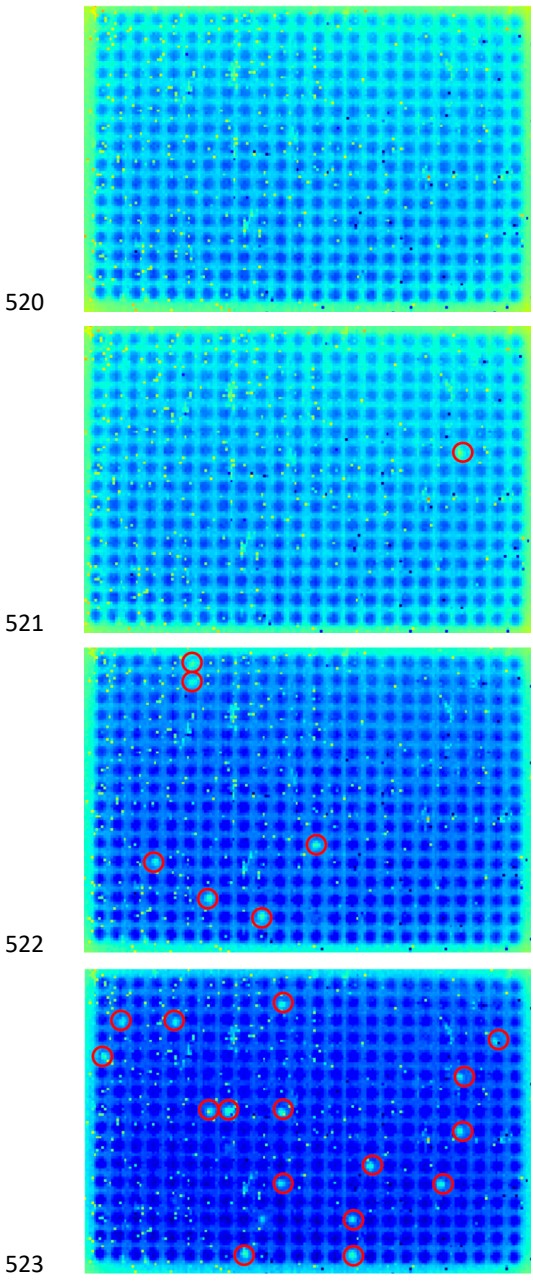

**Figure 4.** Sequence of infrared camera images showing 384 droplets during cooling. Red circles indicate freezing droplets.





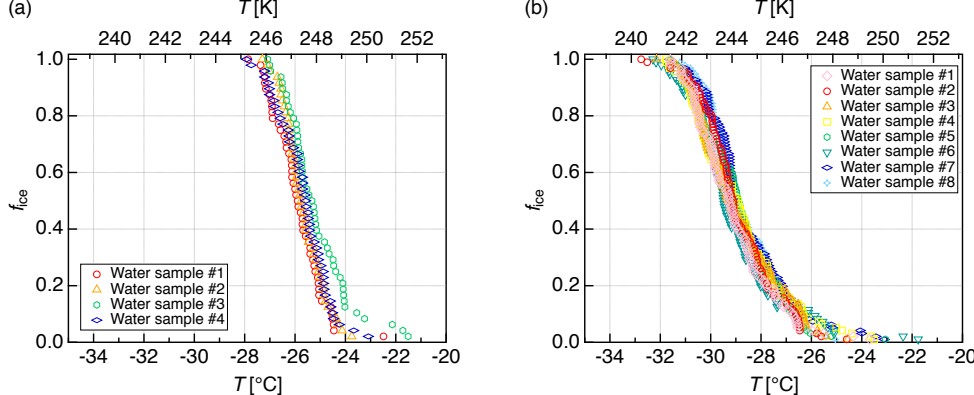

**Figure 5.** Freezing experiments with pure water droplets. Fraction of frozen droplets ($f_{\mathrm{ice}}$) vs. temperature $T$ obtained with a continuous cooling rate of 1 K min$^{-1}$ and a droplet volume of 3 µL. **(a)** Four different samples with 48 droplets each apportioned over two 96-well plates. **(b)** Eight different samples with 96 droplets each apportioned over two 384-well plates.



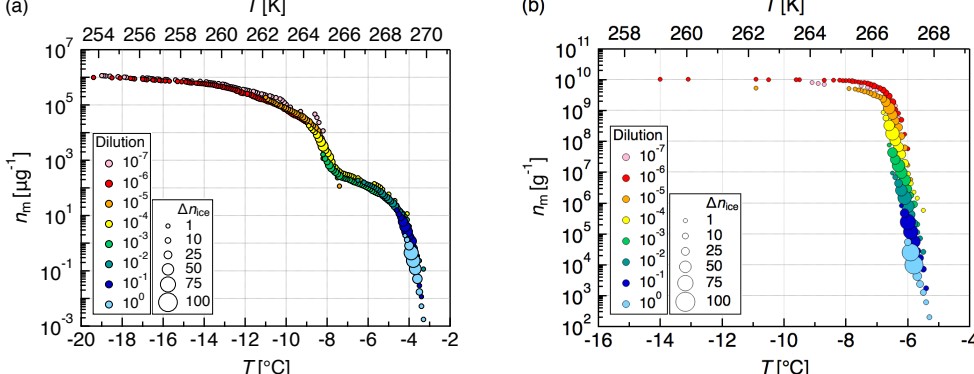

531

**Figure 6.** Measurements of dilution series of **(a)** bacterial IN (Snomax®) and **(b)** fungal IN

(*Mortierella alpina* 13A). Cumulative number of IN ($n_m$) per unit mass of Snomax® and

mycelium, respectively, vs. temperature $T$. Droplets of the same dilution of three independent

measurements were added to a total droplet number of 288 (3 x 96 droplets). Symbol colors

indicate different dilutions; symbol size indicates the number of frozen droplets per temperature

interval ($\Delta T = 0.1$ K).

538





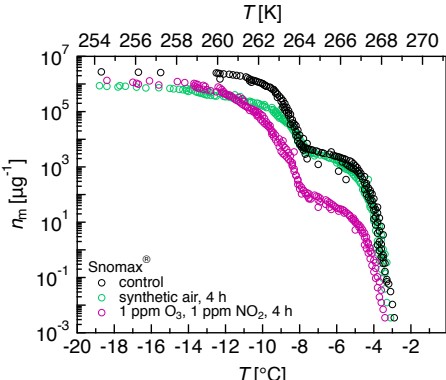

**Figure 7:** Freezing experiments with ozonized and nitrated bacterial IN. Cumulative number of IN ($n_m$) per unit mass of Snomax® vs. temperature $T$. Symbol colors indicate different exposure conditions.





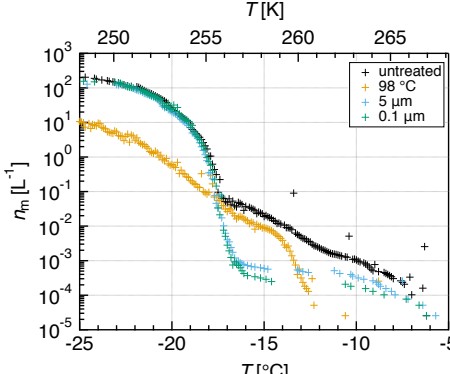

543

**Figure 8.** Freezing experiments with aqueous extracts of atmospheric aerosols. Number of IN

per liter air vs. temperature $T$ for untreated (black), heated (orange), 5 µm filtered (blue), and

0.1 µm filtered (green) filter extracts. Data of different dilutions were averaged for each

treatment.