# Peer review of "Twin-plate ice nucleation assay (TINA) with infrared detection for high-throughput droplet freezing experiments with biological ice nuclei in laboratory and field samples"

_Atmospheric Measurement Techniques, 2018_

## Referee Comment (RC1) · Anonymous Referee #1 · 30 Aug 2018

General Comments:

Kunert et al. designed a new instrument for measuring immersion mode using multi-well plates analyzed simultaneously with a high-throughput experiments. Freezing events are detected automatically based on the heat release during crystallization using an infrared detector. TINA was tested with comprehensive characterization of bacterial (Pseudomonas syringae ,Snomax$^{®}$) and fungal (Mortierella alpine) ice nuclei, as well as atmospheric particles collected on filters. Chemical processing effect of the biological ice nuclei was shown to decrease their activity. The conclusion of

this manuscript is that TINA is suitable for high-throughput experiments for analyzing field and laboratory samples, and characterizing the ice nucleating particles. This manuscript is suited for publication in AMT, and can have valuable contribution to experimentalist in the field of atmospheric ice nucleation. Overall, the manuscript is well structured and I recommend to publish it in AMT after issues raised in this review will be addressed. The authors should stress what is the scientific innovation in their instrument given the very recent paper of Harrison et al. (2018), which was mentioned shortly in the end of the introduction section. Also, why infrared detector enables improved detection over other methods? The ability of high-throughput experiments was mentioned repeatedly in the manuscript, and it will be valuable contribution if the authors could use their existing data to show if this ability is important. I also wonder why error bars are lacking from all data and figures.

Specific comments:

Line #26: It is stated that there is a good agreement with literature data. Where was this shown or detailed in the manuscript?

Line #76: I think it is confusing: up to 10 K min-1 or more?

Line #83: Is this the correct place to introduce the similar approach by Harrison et al.?

Line #94: Here it is not clear if the plates are commercial product or self-designed? If commercial, manufacture details should be specified.

Line #137: It confused me that it was cooled to 218.2 K and heated from 220.7 K?

Section 2.2: So what is the temperature uncertainty of TINA and how was it propagated?

Line 144: I think it is still not clear at this point what is the temperature gradient you refer to. I would first defined that.

Line #151: please clarify why do you mention here Zaragotas et al. (2016).

[Figure]

Line #152: I think it would be nice if you will add the plate temperature at the different images

Line #157: what is the resolution in which images are taken?

Line #182: Are those new plates? or the same plates described earlier in the text?

Line #209: Please add a reference to this claim.

Line #235: Is this correct? Class A only seen for high suspension concentrations.

Line #302: per liter air? Or liter water.

Technical corrections:

Line #97: Fig. 1b should be describes before Fig. 1c.

Line #165: add "is" after Vdrop, and m, and etc..

Line #206: You can remove 'respectively'.

Line #209: 'showed' and not 'show'. Also found in other places in the text.

---

## Referee Comment (RC2) · Anonymous Referee #2 · 2 Sep 2018

Kunert et al. present a method, Twin-plate ice nucleation assay (TINA), for droplet freezing using infrared detection based on the heat release at freezing. Two types of biological INPs and an ambient sample collected on filter were used to test the method. The effect of O3 and NO2 exposure on IN activity of P. syrinagae is also tested. In the manuscript, the authors concluded that this is a high-throughput method which can be applied to lab and field studies. The authors showed that the exposure to high O3 and NO2 could decrease the ice nucleation efficiency of Snomax. From the field sample test, it is concluded that there are efficient biological IN that were smaller than 0.1um.

[Figure]

This manuscript is suitable for publication in AMT. It is recommended to published in AMT after the following issues are addressed.

Comments:

1. Line 21, please be clear what does "deviations <0.5 K" mean and why it is high-precision temperature control.

2. Line 37, it is not clear why this method is suitable for high-throughput experiments and efficient analysis for field samples. I guess the authors don't mean in-situ field measurement? If I understood it right, as for the demonstration of the field sample in the manuscript, one should collect enough sample and extract them carefully to get different dilutions for nm(T) measurements.

3. Line 52-53, why the IR detector is better for droplet freezing detection? Please provide more details.

4. Line 79-80, the two biological INPs are very efficient IN, TINA could work very well with such efficient INPs. How about the situation when the unknown samples are less efficient especially when background freezing start contribute the freezing at e.g., -26C or higher.

5. Line 140, how the droplet temp. is calibrated? you only calibrate the sensors as described? Line 150, the sample temperature needs to be calibrated., not just the sensor. Where are the thermistors in the experimental setup? Depending the location, likely there will be thermal lag due to the plate thickness and different cooling rate. And is the temp. uniform cross such large sample plate? This could also contribute to the different freezing curves for different plates (or plate designs).

6. Line 188, when using cooling rate of 1K/min, does the Wegener-Bergeron-Findeisen process affect the measurement?

7. Line 256-257, please provide a brief description for the O3 and NO2 exposure experiment. Why such high concentration for both O3 and NO2 is used?

8. Line 277, please provide the ice nucleation data for the field blank samples for comparison with ambient sample.

9. Line 298-299, it is not clear why the decrease in IN activity after heat treatment is indication of the presence of biological IN.

10. Line 307-314, it is suggested to carefully evaluate the discussion and conclusion in this section. As showing in Fig. 8, there is less than one order of magnitude different between the samples after 5 um and 0.1 um filtration. This could be just within measurement uncertainty, which is showing in Fig. S1 and S2, for each three independent samples that there is about one order of magnitude variation at certain temperature range.

11. Fig. 4, please provide the detail description for freezing determination using IR camera.

---

## Author Response (AR1)

**MS amt-2018-230, Kunert et al.: Twin-plate ice nucleation assay (TINA) with infrared detection for high-throughput droplet freezing experiments with biological ice nuclei in laboratory and field samples**

We thank referee #1 for his comments, questions, and suggestions, which are highly appreciated and have been taken into account upon revision of our manuscript. The comments and our answers are listed below.

Referee comment: The authors should stress what is the scientific innovation in their instrument given the very recent paper of Harrison et al. (2018), which was mentioned shortly in the end of the introduction section.
Author's response: TINA studies ice nucleation in 960 microliter range droplets in one experiment, which enables the analysis of many samples or dilution series with good statistics in a short period of time. The cooling system allows a fast and high-precision temperature control over a wide temperature range at variable cooling rates. The infrared detection is an efficient method to doubtlessly determine freezing events, which was first applied to droplet freezing assays by Zaragotas et al. 2016. As discussed by Grothe 2018 (doi:10.5194/amt-2018-177-RC3, 2018), the authors of Harrison et al. 2018 attended several workshops and conferences and also have been the organizers for one conference, where our Twin-plate ice nucleation assay (TINA) with infrared detection was presented and discussed, so that our new setup was well-known to them.

Referee comment: Also, why infrared detector enables improved detection over other methods?
Author's response: The infrared detector monitors the temperature of each droplet during cooling. As soon as a droplet freezes, latent heat is released and a sharp signal can be detected. For clarification, we modified the last sentence of paragraph 2 in section 1, where we replaced "improved" by "efficient".

Referee comment: The ability of high-throughput experiments was mentioned repeatedly in the manuscript, and it will be valuable contribution if the authors could use their existing data to show if this ability is important.
Author's response: TINA is suitable for high-throughput experiments because the instrument enables the study of ice nucleation in 960 microliter range droplets in one experiment, which enables the analysis of many samples or dilution series with good statistics in a short period of time. This is demonstrated in Figure 10, for which aqueous extracts of two aliquots of an atmospheric aerosol filter sample were treated in three different ways. All treated samples and untreated controls were measured in five different dilutions to provide the full ice nucleation spectrum for each sample. Each dilution was measured in 96 droplets, and every sample consisted of two aliquots. All in all, 4608 droplets were measured for Figure 10, which correspond to six experiments performed by TINA. For each freezing experiment down to -30 °C, TINA takes about 45 min, which means 4.5 h of operation of TINA for Figure 10.

Referee comment: I also wonder why error bars are lacking from all data and figures.
Author's response: The uncertainty of the temperature sensor was used as the error of the temperature and was added into the figure captions. The error of the IN number concentrations was calculated using the counting error and the Gaussian error propagation, and error bars of the IN number concentrations were added into all figures.

The section 2.4 was optimized and the following paragraph was included:

"Assuming ice nucleation as a time-independent (singular) process, the number concentration
of IN ($\frac{\Delta N_m}{\Delta T}$) active at a certain temperature ($T$) per unit mass of material is given by Eq. (1)
(Vali, 1971a).

$\quad \frac{\Delta N_m}{\Delta T}(T) = -\ln\left(1 - \frac{s}{a - \Sigma_{i=0}^{j} s}\right) \cdot \frac{c}{\Delta T} \qquad\qquad ; 0 \leq j \leq a$ (1)

with $c = \frac{V_{\text{wash}}}{V_{\text{drop}}} \cdot \frac{d}{m}$ (2)

where $s$ is the number of freezing events in 0.1 K bins ($\Delta T$), $a$ is the number of all droplets, $m$
is the mass of the particles in the initial suspension, $V_{\text{wash}}$ is the volume of the initial suspension,
$V_{\text{drop}}$ is the droplet volume, and $d$ is the dilution factor of the droplets relative to $m$. The
measurement uncertainty ($\delta\frac{\Delta N_m}{\Delta T}(T)$) was calculated using the counting error of $s$ plus one digit
and the Gaussian error propagation (Eq. (3)).

$\quad \delta\frac{\Delta N_m}{\Delta T}(T) = \sqrt{\left(\frac{1}{1 - \frac{s}{a - \Sigma_{i=0}^{j} s}} \cdot \frac{c}{\Delta T} \cdot \frac{\sqrt{s+1}}{a - \Sigma_{i=0}^{j} s}\right)^2 + \left(\frac{1}{1 - \frac{s}{a - \Sigma_{i=0}^{j} s}} \cdot \frac{c}{\Delta T} \cdot \frac{s \cdot \sqrt{\Sigma_{i=0}^{j} s+1}}{\left(a - \Sigma_{i=0}^{j} s\right)^2}\right)^2}$ (3)

The cumulative IN number concentration ($N_m(T)$) is given by Eq. (4).

$\quad N_m(T) = -\ln\left(1 - \frac{\Sigma_{i=0}^{j} s}{a}\right) \cdot c \qquad\qquad ; 0 \leq j \leq a$ (4)

The error of the cumulative IN number concentration ($\delta N_m(T)$) was calculated using Eq. (5).

$\quad \delta N_m(T) = \sqrt{\left(\frac{c}{1 - \frac{\Sigma_{i=0}^{j} s}{a}} \cdot \frac{\sqrt{\Sigma_{i=0}^{j} s+1}}{a}\right)^2}$ (5)

According to the above equations, the uncertainty is proportional to the number of frozen
droplets per temperature bin. In the freezing experiments described below, the lowest number
of freezing events and largest uncertainties were obtained at the lower and higher end of each
dilution series (Poisson distribution). Data points with uncertainties ≥100% were excluded
(overall less than 6% of the measurement data)."
Specific comments:
Referee comment: Line #26: It is stated that there is a good agreement with literature data.
Where was this shown or detailed in the manuscript?
Author's response: In section 3.2, we discussed the results of our experiments with Snomax®,
which are shown in Figures 7 and S4. "These findings are in accordance with the results of
Budke and Koop (2015)." Here, we replaced "in accordance" with "in good agreement".
In the same section, we also discussed the results of our experiments with *Mortierella alpina*,
which are shown in Figures 8 and S5. "The cumulative number of IN and the initial freezing
temperature of 268 K (-5 °C) are in good agreement with the literature (Fröhlich-Nowoisky et
al., 2015; Pummer et al., 2015)."
Referee comment: Line #76: I think it is confusing: up to 10 K min-1 or more?
Author's response: We tested our setup with continuous cooling rates of up to 10 K min$^{-1}$, but
it is possible to run the setup at higher cooling rates. But it has to be considered, that for each
cooling rate a new correction matrix has to be generated. For clarification, we deleted "or
more".

Author's response: We deleted the sentence "Very recently, a similar approach for droplet freezing experiments with IR detection has been presented by Harrison et al. (2018), investigating K-feldspar, NX-illite, and atmospheric aerosol samples." at the end of section 1, and we modified in section 1: "Infrared (IR) detectors enable efficient detection of droplet freezing (Harrison et al., 2018; Zaragotas et al., 2016)."

Author's response: We added the following sentence: "For each experiment, new sterile multiwell plates are used (96-well: Axon Labortechnik Kaiserslautern, Germany, 384-well: Eppendorf, Hamburg, Germany)."

Author's response: To clarify this, we modified the sentence: "The temperature within the bath was cooled down from 303.2 K to 218.2 K (30.0 °C to -55.0 °C) in 5 K steps, warmed to 220.7 K (-52.5 °C), and raised again from 220.7 K to 300.7 K (-52.5 °C to 27.5 °C) in 5 K steps."

Author's response: The temperature uncertainty of TINA is 0.2 K. We added the following sentence at the end of section 2.2: "From the calibration measurements, we obtained a total uncertainty estimate of $\delta_{total} < 0.2$ K ($\delta_{total} = \delta_{Thermistor} + \delta_{TC} + \delta_{Morti}$)."

Author's response: We included thermocouple measurements in the individual wells of the sample holder blocks to correct for a temperature gradient within the two blocks. We added the following paragraph at the end of section 2.2. and included four new figures (Figure 4, S1, S2, S3) while renaming the existing. "To determine a potential temperature gradient of the sample holder blocks, two thermocouples (K type, 0.08 mm diameter, Omega) were positioned in various wells of multiwell plates (Figure S1a/b), each filled with 30 µL pure water (see Sect. 3.1). These thermocouples were connected to the thermocouple in the elevation of each sample holder block, and the temperature offset between sample holder block and wells was measured for a continuous cooling rate of 1 K min$^{-1}$ (Figure S1c). Below -2 °C, the temperature offset between sample holder block and wells is nearly constant, in this example ~0.16 K and ~0.19 K. The measurement was performed in duplicates for all observed wells. Figure S2 shows the temperature gradient exemplarily for the 384-well sample holder block in a 2D interpolation based on all measurements.
To characterize the uncertainty of this measurement, the two thermocouples were placed in an ice water bath, and the sample holder block was cooled down to 2 °C, 1 °C, 0 °C, -1 °C, and -2 °C ($T_{block}$), while the difference between the ice water and the block temperature was monitored by the thermocouples ($T_{diffTC}$) (Figure S3). From these experiments, we obtained thermocouple uncertainties $\delta_{TC} < 0.05$ K ($\delta_{TC} = T_{block} + T_{diffTC}$).
Additionally, we used undiluted IN filtrate of *Mortierella alpina* 13A (see Sect. 3.2) as calibration substance, and a freezing experiment was performed as described for the biological reference materials (see Sect. 3.2). These results were used to compensate for the temperature gradient, and the thermocouple measurements were used to correct the temperature offset between gradient-corrected wells and thermistors. A correction matrix was calculated, and this matrix was used to correct subsequent freezing experiments. Figure 4 shows the results of the fungal IN filtrate measurement (a) before and (b) after correction. After correction, all fungal IN filtrate measurements showed a standard deviation of $< 0.06$ K ($\delta_{\text{Morti}}$). From the calibration measurements, we obtained a total uncertainty estimate of $\delta_{\text{total}} < 0.2$ K ($\delta_{\text{total}} = \delta_{\text{Thermistor}} + \delta_{\text{TC}} + \delta_{\text{Morti}}$)."

Referee comment: Line #151: please clarify why do you mention here Zaragotas et al. (2016).
Author's response: We deleted the sentence "In contrast, Zaragotas et al. (2016) used infrared camera, which was calibrated only once by the company, to measure the accurate temperature of each droplet."

Referee comment: Line #152: I think it would be nice if you will add the plate temperature at the different images.
Author's response: We thank the referee for this suggestion, and we added the plate temperature at the different images.

Referee comment: Line #157: what is the resolution in which images are taken?
Author's response: We added the information about the resolution of the images to section 2.3 to specify the method: "The camera has a resolution of 206 x 156 pixels, and it takes ten pictures per second. These pictures are averaged to one picture per second."

Referee comment: Line #182: Are those new plates? or the same plates described earlier in the text?
Author's response: We changed the sentence as follows: "For background measurements, 3 μL aliquots of autoclaved and filtered pure water were pipetted into new sterile multiwell plates by a liquid handling station."
Moreover, we added this information in section 2.1.: "For each experiment, new sterile multiwell plates are used (96-well: Axon Labortechnik Kaiserslautern, Germany, 384-well: Eppendorf, Hamburg, Germany)."

Referee comment: Line #209: Please add a reference to this claim.
Author's response: We assume that different plates from different manufactures can lead to differences in freezing because of the production process and well shape, but cannot add a specific reference. For clarification we changed the sentence as follows: "The 96-well plates were obtained from a different manufacturer than the 384-well plates."

Referee comment: Line #235: Is this correct? Class A only seen for high suspension concentrations.
Author's response: We thank the referee for this comment. We removed the following text: "These differences result from three different classes of IN with different activation temperatures as described by Turner et al. (1990). Based on this classification, the Snomax® sample contains a large number of class A and C IN, but only a few IN of class B. These findings are in accordance with the results of Budke and Koop (2015). Below 259 K (-14 °C), a flat plateau arises where no IN are active." and we included the following sentence: "These findings are in good agreement with the results of Budke and Koop (2015)"

Referee comment: Line #302: per liter air? Or liter water.

Author's response: The IN concentration was calculated per liter air, which passed the filter
during sampling. We included the following sentence: "All IN concentrations were calculated
per liter air."
Technical corrections:
Referee comment: Line #97: Fig. 1b should be describes before Fig. 1c.
Author's response: Changed as suggested.
Referee comment: Line #165: add "is" after Vdrop, and m, and etc..
Author's response: We modified the sentence.
Referee comment: Line #206: You can remove 'respectively'.
Author's response: This has been removed.
Referee comment: Line #209: 'showed' and not 'show'. Also found in other places in the text.
Author's response: We replaced it in several places in the text.

**MS amt-2018-230, Kunert et al.: Twin-plate ice nucleation assay (TINA) with infrared detection for high-throughput droplet freezing experiments with biological ice nuclei in laboratory and field samples**

We thank referee #2 for his comments, questions, and suggestions, which are highly appreciated and have been taken into account upon revision of our manuscript. The comments and our responses are listed below.

Referee comment 1: Line 21, please be clear what does "deviations <0.5 K" mean and why it is high-precision temperature control.

Author's response: We recalculated our maximum total error and replaced "deviations < 0.5 K" by "uncertainty < 0.2 K". We included a new Figure S3, which shows the high-precision temperature control, by which TINA is operated.

Referee comment 2: Line 37, it is not clear why this method is suitable for high-throughput experiments and efficient analysis for field samples. I guess the authors don't mean in-situ field measurement? If I understood it right, as for the demonstration of the field sample in the manuscript, one should collect enough sample and extract them carefully to get different dilutions for nm(T) measurements.

Author's response: TINA is suitable for high-throughput experiments because the instrument enables the study of ice nucleation in 960 microliter range droplets in one experiment, which enables the analysis of many samples or dilution series with good statistics in a short period of time. This is demonstrated in Figure 10, for which aqueous extracts of two aliquots of an atmospheric aerosol filter sample were treated in three different ways. All treated samples and untreated controls were measured in six different dilutions to provide the full ice nucleation spectrum for each sample. Each dilution was measured in 96 droplets. All in all, 4608 droplets were measured for Figure 10, which correspond to six experiments performed by TINA. For each freezing experiment down to -30 °C, TINA takes about 45 min, which means 4.5 h of operation of TINA for Figure 10.

Referee comment 3: Line 52-53, why the IR detector is better for droplet freezing detection? Please provide more details.

Author's response: The infrared camera monitors the temperature of each droplet during cooling. As soon as a droplet freezes, latent heat is released and a sharp signal can be detected. We modified the last sentence of paragraph 2 in section 1, where we replaced "improved" by "efficient".

Referee comment 4: Line 79-80, the two biological INPs are very efficient IN, TINA could work very well with such efficient INPs. How about the situation when the unknown samples are less efficient especially when background freezing start contribute the freezing at e.g., -26C or higher.

Author's response: Our experiments with aqueous extracts of atmospheric aerosols confirm that TINA is suitable for freezing experiments of samples with unknown IN in a temperature range down to -25 °C. Below this temperature, background freezing needs to be considered.

Referee comment 5: Line 140, how the droplet temp. is calibrated? you only calibrate the sensors as described? Line 150, the sample temperature needs to be calibrated., not just the sensor. Where are the thermistors in the experimental setup? Depending the location, likely there will be thermal lag due to the plate thickness and different cooling rate. And is the temp. uniform cross such large sample plate? This could also contribute to the different freezing curves for different plates (or plate designs).

Author's response: We included thermocouple measurements in the individual wells of the
sample holder blocks to correct for a temperature gradient within the two blocks. We added the
following paragraph at the end of section 2.2. and included four new figures (Figure 4, S1, S2,
S3) while renaming the existing. "To determine a potential temperature gradient of the sample
holder blocks, two thermocouples (K type, 0.08 mm diameter, Omega) were positioned in
various wells of multiwell plates (Figure S1a/b), each filled with 30 µL pure water (see Sect.
3.1). These thermocouples were connected to the thermocouple in the elevation of each sample
holder block, and the temperature offset between sample holder block and wells was measured
for a continuous cooling rate of 1 K min$^{-1}$ (Figure S1c). Below -2 °C, the temperature offset
between sample holder block and wells is nearly constant, in this example ~0.16 K and ~0.19
K. The measurement was performed in duplicates for all observed wells. Figure S2 shows the
temperature gradient exemplarily for the 384-well sample holder block in a 2D interpolation
based on all measurements.
To characterize the uncertainty of this measurement, the two thermocouples were placed in an
ice water bath, and the sample holder block was cooled down to 2 °C, 1 °C, 0 °C, -1 °C, and -
2 °C ($T_{block}$), while the difference between the ice water and the block temperature was
monitored by the thermocouples ($T_{diffTC}$) (Figure S3). From these experiments, we obtained
thermocouple uncertainties $\delta_{TC} < 0.05$ K ($\delta_{TC} = T_{block} + T_{diffTC}$).
Additionally, we used undiluted IN filtrate of *Mortierella alpina* 13A (see Sect. 3.2) as
calibration substance, and a freezing experiment was performed as described for the biological
reference materials (see Sect. 3.2). These results were used to compensate for the temperature
gradient, and the thermocouple measurements were used to correct the temperature offset
between gradient-corrected wells and thermistors. A correction matrix was calculated, and this
matrix was used to correct subsequent freezing experiments. Figure 4 shows the results of the
fungal IN filtrate measurement (a) before and (b) after correction. After correction, all fungal
IN filtrate measurements showed a standard deviation of $< 0.06$ K ($\delta_{Morti}$). From the calibration
measurements, we obtained a total uncertainty estimate of $\delta_{total} < 0.2$ K ($\delta_{total} = \delta_{Thermistor} + \delta_{TC}$
$+ \delta_{Morti}$)."
Referee comment 6: Line 188, when using cooling rate of 1K/min, does the Wegener-Bergeron-
Findeisen process affect the measurement?
Author's response: The individual droplets are in separate compartments and do not influence
each other during the freezing experiment.
Referee comment 7: Line 256-257, please provide a brief description for the O3 and NO2
exposure experiment. Why such high concentration for both O3 and NO2 is used?
Author's response: In section 3.3, we included the following paragraph: "Briefly, $O_3$ was
produced by exposing synthetic air to UV light (L.O.T.-Oriel GmbH & Co. KG, Germany), and
the $O_3$ concentration was adjusted by tuning the amount of UV light. The gas flow was ~1.9 L
min$^{-1}$, and it was mixed with $N_2$ containing ~5 ppmV $NO_2$ (Air Liquide, Germany). The $NO_2$
concentration was regulated by the addition of the amount of the ~5 ppmV $NO_2$ gas. The $O_3$
and $NO_2$ concentrations were monitored with commercial monitoring instruments (ozone
analyzer: 49i, Thermo Scientific, Germany; $NO_x$ analyzer: 42i-TL, Thermo Scientific). The gas
mixture was directly bubbled through 1 mL of the Snomax® solution at a flow rate of 60 mL
min$^{-1}$ using a Teflon tube (ID: 1.59 mm). The Snomax® solution was exposed to a mixture of 1
ppm $O_3$ and 1 ppm $NO_2$ for 4 h, representing the exposure to an atmospherically relevant
amount of about 200 ppb each for about 20 h. The exposure experiments were performed in
triplicates. After exposure, the treated samples were serially diluted and the IN activity was
measured as described for the Snomax® reference measurements."

Referee comment 8: Line 277, please provide the ice nucleation data for the field blank samples
for comparison with ambient sample.
Author's response: We included the data for the blank sample in Figure 10.
Referee comment 9: Line 298-299, it is not clear why the decrease in IN activity after heat
treatment is indication of the presence of biological IN.
Author's response: For clarification, we added the following sentence: "The concentration of
IN active at temperatures above 263 K (-10 °C) was about 0.001 L$^{-1}$, but heat treatment led to
a loss of IN activity above 263 K (-10 °C). Because the activity of known biological IN results
from proteins or proteinaceous compounds (Green and Warren, 1985; Kieft and Ruscetti, 1990;
Pouleur et al., 1992; Tsumuki and Konno, 1994), and proteins are known to be heat-sensitive,
the results suggest the presence of biological IN."
Referee comment 10: Line 307-314, it is suggested to carefully evaluate the discussion and
conclusion in this section. As showing in Fig. 8, there is less than one order of magnitude
different between the samples after 5 um and 0.1 um filtration. This could be just within
measurement uncertainty, which is showing in Fig. S1 and S2, for each three independent
samples that there is about one order of magnitude variation at certain temperature range.
Author's response: We thank the referee for his suggestion. We carefully evaluated the results
and rewrote the paragraph: "Filtration experiments did not affect the initial freezing
temperature, but the concentration of biological IN decreased significantly. The results suggest
the presence of many biological IN or agglomerates larger than 5 μm and of a few biological
IN smaller than 0.1 μm. The cumulative number of IN active between 263 K (-10 °C) and 257
K (-16 °C) decreased up to two orders of magnitude upon filtration, but the IN concentration
below 256 K (-17 °C) was not affected. The findings show that many IN active between 263 K
(-10 °C) and 257 K (-16 °C) were larger than 5 μm, whereas IN active below 256 K (-17 °C)
were smaller than 0.1 μm."
Referee comment 11: Fig. 4, please provide the detail description for freezing determination
using IR camera.
Author's response: The infrared camera monitors the temperature of each droplet during
cooling. As soon as a droplet freezes, latent heat is released and a sharp signal can be detected.
At the end of section 2.3, the freezing determination using IR camera is explained: "Software
analysis uses a grid of 96 and 384 points, respectively, where the grid point is set to the center
of each well enabling to fit the dimensions of each plate under different perspective angles. The
temperature is tracked for each well during the experiment. A self-written algorithm detects a
local maximum shortly followed by a local minimum in the derivative of the temperature
profile, which is caused by the release of latent heat during freezing. The software exports the
data for each droplet in CSV format."
Additionally, we added the information about the resolution of the images to section 2.3 to
specify the method: "The camera has a resolution of 206 x 156 pixels, and it takes ten pictures
per second. These pictures are averaged to one picture per second."

 **List of changes**
All changes have been marked in the revised version of the manuscript using track-changes.
The most relevant changes are listed below.

Author list:
The order was changed from "Anna T. Kunert[1], Mark Lamneck[2], Frank Helleis[2], Mira L.
Pöhlker[1], Ulrich Pöschl[1], and Janine Fröhlich-Nowoisky[1,*]" to "Anna T. Kunert[1], Mark
Lamneck[2], Frank Helleis[2], Ulrich Pöschl[1], Mira L. Pöhlker[1,*], and Janine Fröhlich-
Nowoisky[1,*]"
The following author was added to the corresponding authors:
Mira L. Pöhlker

Figures:
All figures of the manuscript and the supplementary material containing experimental data have
been modified regarding data processing and error bars. Figure 4 was added to the manuscript.
Figure 6a and 6b have been divided into two separate figures (Figure 7a and 8a), and differential
IN spectra were added (Figure 7b and 8b). All subsequent figures have been renamed. In the
supplementary material, the following figures have been added: S1, S2, S3, while renaming the
existing.

Section 2.2:
We included thermocouple measurements in the individual wells of the sample holder blocks
to correct for a temperature gradient within the two blocks. We added the following paragraph
at the end of section 2.2. and included four new figures (Figure 4, S1, S2, S3) while renaming
the existing.
"To determine a potential temperature gradient of the sample holder blocks, two thermocouples
(K type, 0.08 mm diameter, Omega) were positioned in various wells of multiwell plates
(Figure S1a/b), each filled with 30 µL pure water (see Sect. 3.1). These thermocouples were
connected to the thermocouple in the elevation of each sample holder block, and the
temperature offset between sample holder block and wells was measured for a continuous
cooling rate of 1 K min$^{-1}$ (Figure S1c). Below -2 °C, the temperature offset between sample
holder block and wells is nearly constant, in this example ~0.16 K and ~0.19 K. The
measurement was performed in duplicates for all observed wells. Figure S2 shows the
temperature gradient exemplarily for the 384-well sample holder block in a 2D interpolation
based on all measurements.
To characterize the uncertainty of this measurement, the two thermocouples were placed in an
ice water bath, and the sample holder block was cooled down to 2 °C, 1 °C, 0 °C, -1 °C, and -
2 °C ($T_{block}$), while the difference between the ice water and the block temperature was
monitored by the thermocouples ($T_{diffTC}$) (Figure S3). From these experiments, we obtained
thermocouple uncertainties $\delta_{TC} < 0.05$ K ($\delta_{TC} = T_{block} + T_{diffTC}$).
Additionally, we used undiluted IN filtrate of *Mortierella alpina* 13A (see Sect. 3.2) as
calibration substance, and a freezing experiment was performed as described for the biological
reference materials (see Sect. 3.2). These results were used to compensate for the temperature
gradient, and the thermocouple measurements were used to correct the temperature offset
between gradient-corrected wells and thermistors. A correction matrix was calculated, and this matrix was used to correct subsequent freezing experiments. Figure 4 shows the results of the
fungal IN filtrate measurement (a) before and (b) after correction. After correction, all fungal
IN filtrate measurements showed a standard deviation of < 0.06 K ($\delta_{Morti}$). From the calibration
measurements, we obtained a total uncertainty estimate of $\delta_{total} < 0.2$ K ($\delta_{total} = \delta_{Thermistor} + \delta_{TC}$
$+ \delta_{Morti}$)."

Section 2.4:
As requested by the referees, we included a detailed error calculation and added error bars to
all figures. The error of the IN number concentrations was calculated using the counting error
and the Gaussian error propagation. The total uncertainty estimate was used as the error of the
temperature and was added into the figure captions. Additionally, we calculated the differential
IN number concentration and plotted the corresponding spectrum.

The former section 2.4. was replaced by "Assuming ice nucleation as a time-independent
(singular) process, the number concentration of IN ($\frac{\Delta N_m}{\Delta T}$) active at a certain temperature ($T$) per
unit mass of material is given by Eq. (1) (Vali, 1971a).

$$\frac{\Delta N_m}{\Delta T}(T) = -\ln\left(1 - \frac{s}{a - \Sigma_{i=0}^{j} s}\right) \cdot \frac{c}{\Delta T} \qquad ; 0 \leq j \leq a \tag{1}$$

with $c = \frac{V_{wash}}{V_{drop}} \cdot \frac{d}{m}$ $\tag{2}$

where $s$ is the number of freezing events in 0.1 K bins ($\Delta T$), $a$ is the number of all droplets, $m$
is the mass of the particles in the initial suspension, $V_{wash}$ is the volume of the initial suspension,
$V_{drop}$ is the droplet volume, and $d$ is the dilution factor of the droplets relative to $m$. The
measurement uncertainty ($\delta\frac{\Delta N_m}{\Delta T}(T)$) was calculated using the counting error of $s$ plus one digit
and the Gaussian error propagation (Eq. (3)).

$$\delta\frac{\Delta N_m}{\Delta T}(T) = \sqrt{\left(\frac{1}{1 - \frac{s}{a - \Sigma_{i=0}^{j} s}} \cdot \frac{c}{\Delta T} \cdot \frac{\sqrt{s+1}}{a - \Sigma_{i=0}^{j} s}\right)^2 + \left(\frac{1}{1 - \frac{s}{a - \Sigma_{i=0}^{j} s}} \cdot \frac{c}{\Delta T} \cdot \frac{s \cdot \sqrt{\Sigma_{i=0}^{j} s + 1}}{\left(a - \Sigma_{i=0}^{j} s\right)^2}\right)^2} \tag{3}$$

The cumulative IN number concentration ($N_m(T)$) is given by Eq. (4).

$$N_m(T) = -\ln\left(1 - \frac{\Sigma_{i=0}^{j} s}{a}\right) \cdot c \qquad ; 0 \leq j \leq a \tag{4}$$

The error of the cumulative IN number concentration ($\delta N_m(T)$) was calculated using Eq. (5).

$$\delta N_m(T) = \sqrt{\left(\frac{c}{1 - \frac{\Sigma_{i=0}^{j} s}{a}} \cdot \frac{\sqrt{\Sigma_{i=0}^{j} s + 1}}{a}\right)^2} \tag{5}$$

According to the above equations, the uncertainty is proportional to the number of frozen
droplets per temperature bin. In the freezing experiments described below, the lowest number
of freezing events and largest uncertainties were obtained at the lower and higher end of each
dilution series (Poisson distribution). Data points with uncertainties $\geq 100\%$ were excluded
(overall less than 6% of the measurement data)."

Section 3.2:
We added additional differential IN spectra for the reference materials and rewrote the
discussion:

"Three independent experiments with Snomax® showed reproducible results (Fig. S4), and,
therefore, droplets of the same dilution were added to a total droplet number of 288. The
obtained results were plotted in a cumulative and a differential IN spectrum (Fig. 7). The
cumulative IN number concentration represents the total number of IN active above a certain
temperature. The cumulative IN spectrum showed two strong increases, around 270 K (-3 °C)
and around 265 K (-8 °C). These findings are in good agreement with the results of Budke and
Koop (2015). The differential IN number concentration was calculated according to Vali
(1971a), and it represents the number of IN active in a particular temperature interval. The
differential IN spectrum showed a similar shape as the cumulative IN spectrum with a distinct
plateau between 268 K and 266 K (-5 °C and -7 °C) and two slight maxima, around 269 K (-4
°C) and around 264 K (-9 °C). This indicates the presence of highly-efficient IN, active at a
temperature of approximately 269 K (-4 °C), and less-efficient IN, active around 264 K (-9 °C).
The fact that the less-efficient IN appeared in higher dilutions implies that they occur in higher
concentrations than the highly-efficient IN. The presence of further IN with lower freezing
temperatures and low concentrations cannot be excluded."
"The cumulative and the differential IN spectra showed similar shapes with one maximum
around 267 K (-6 °C), indicating the presence of one type of IN, which is highly-efficient."
Section 3.3:
We included the following paragraph to provide a brief description for the $O_3$ and $NO_2$ exposure
experiments:
"Briefly, $O_3$ was produced by exposing synthetic air to UV light (L.O.T.-Oriel GmbH & Co.
KG, Germany), and the $O_3$ concentration was adjusted by tuning the amount of UV light. The
gas flow was ~1.9 L min$^{-1}$, and it was mixed with $N_2$ containing ~5 ppmV $NO_2$ (Air Liquide,
Germany). The $NO_2$ concentration was regulated by the addition of the amount of the ~5 ppmV
$NO_2$ gas. The $O_3$ and $NO_2$ concentrations were monitored with commercial monitoring
instruments (ozone analyzer: 49i, Thermo Scientific, Germany; $NO_x$ analyzer: 42i-TL, Thermo
Scientific). The gas mixture was directly bubbled through 1 mL of the Snomax® solution at a
flow rate of 60 mL min$^{-1}$ using a Teflon tube (ID: 1.59 mm). The Snomax® solution was
exposed to a mixture of 1 ppm $O_3$ and 1 ppm $NO_2$ for 4 h, representing the exposure to an
atmospherically relevant amount of about 200 ppb each for about 20 h. The exposure
experiments were performed in triplicates. After exposure, the treated samples were serially
diluted and the IN activity was measured as described for the Snomax® reference
measurements."
We rewrote the results:
"The results showed that gas exposure affected the IN activity of Snomax® (Fig. 9).  High
concentrations of $O_3$ and $NO_2$ reduced the cumulative number of IN from Snomax® between
one and two orders of magnitude, while exposure to synthetic air showed smaller effects."
Section 3.4:
We carefully evaluated the results and rewrote the paragraph:
"Filtration experiments did not affect the initial freezing temperature, but the concentration of
biological IN decreased significantly. The results suggest the presence of many biological IN
or agglomerates larger than 5 μm and of a few biological IN smaller than 0.1 μm. The cumulative number of IN active between 263 K (-10 °C) and 257 K (-16 °C) decreased up to
two orders of magnitude upon filtration, but the IN concentration below 256 K (-17 °C) was
not affected. The findings show that many IN active between 263 K (-10 °C) and 257 K (-16
°C) were larger than 5 µm, whereas IN active below 256 K (-17 °C) were smaller than 0.1 µm."
We removed the following paragraph from the manuscript, because the outliers were excluded
based on the introduced error calculation.

[revised manuscript text omitted]

$$\text{with } c = \frac{V_{wash}}{V_{drop}} \cdot \frac{d}{m} \qquad (2)$$

where $s$ is the number of freezing events in 0.1 K bins ($\Delta T$), $a$ is the number of all droplets, $m$
is the mass of the particles in the initial suspension, $V_{wash}$ is the volume of the initial suspension,
$V_{drop}$ is the droplet volume, and $d$ is the dilution factor of the droplets relative to $m$. The
measurement uncertainty ($\delta \frac{\Delta N_m}{\Delta T}(T)$) was calculated using the counting error of $s$ plus one digit
and the Gaussian error propagation (Eq. (3)).

$$\delta \frac{\Delta N_m}{\Delta T}(T) = \sqrt{\left(\frac{1}{1 - \frac{s}{a - \sum_{i=0}^{j} s}} \cdot \frac{c}{\Delta T} \cdot \frac{\sqrt{s+1}}{a - \sum_{i=0}^{j} s}\right)^2 + \left(\frac{1}{1 - \frac{s}{a - \sum_{i=0}^{j} s}} \cdot \frac{c}{\Delta T} \cdot \frac{s \cdot \sqrt{\sum_{i=0}^{j} s + 1}}{\left(a - \sum_{i=0}^{j} s\right)^2}\right)^2} \qquad (3)$$

The cumulative IN number concentration ($N_m(T)$) is given by Eq. (4).

$$N_m(T) = -\ln\left(1 - \frac{\sum_{i=0}^{j} s}{a}\right) \cdot c \qquad ; 0 \leq j \leq a \qquad (4)$$

The error of the cumulative IN number concentration ($\delta N_m(T)$) was calculated using Eq. (5).

$n_m(T) = \frac{-\ln(1 - f_{ice})}{V_{drop}} \cdot \frac{d}{m}$ (1)

[revised manuscript text omitted]

[2] nach oben verschoben: **Figure 8.**